# The Role of HGF/MET Signaling in Metastatic Uveal Melanoma

**DOI:** 10.3390/cancers13215457

**Published:** 2021-10-30

**Authors:** Ryota Tanaka, Mizue Terai, Eric Londin, Takami Sato

**Affiliations:** 1Department of Medical Oncology, Sidney Kimmel Cancer Center, Thomas Jefferson University, Philadelphia, PA 19107, USA; ryota.tanaka@jefferson.edu (R.T.); takami.sato@jefferson.edu (T.S.); 2Department of Pathology, Anatomy and Cell Biology, Thomas Jefferson University, Philadelphia, PA 19107, USA; eric.londin@jefferson.edu

**Keywords:** hepatocyte growth factor (HGF), mesenchymal-epithelial transition factor (MET), uveal melanoma, metastasis

## Abstract

**Simple Summary:**

Hepatocyte growth factor (HGF)/mesenchymal-epithelial transition factor (MET) signaling plays an important role in the metastatic formation and therapeutic resistance to uveal melanoma. Here, we review the various functions of MET signaling contributing to metastatic formation, as well as review resistance to treatments in metastatic uveal melanoma.

**Abstract:**

Hepatocyte growth factor (HGF)/mesenchymal-epithelial transition factor (MET) signaling promotes tumorigenesis and tumor progression in various types of cancer, including uveal melanoma (UM). The roles of HGF/MET signaling have been studied in cell survival, proliferation, cell motility, and migration. Furthermore, HGF/MET signaling has emerged as a critical player not only in the tumor itself but also in the tumor microenvironment. Expression of MET is frequently observed in metastatic uveal melanoma and is associated with poor prognosis. It has been reported that HGF/MET signaling pathway activation is the major mechanism of treatment resistance in metastatic UM (MUM). To achieve maximal therapeutic benefit in MUM patients, it is important to understand how MET signaling drives cellular functions in uveal melanoma cells. Here, we review the HGF/MET signaling biology and the role of HGF/MET blockades in uveal melanoma.

## 1. Introduction

Uveal melanoma (UM), which arises from melanocytes in the choroid (85–90%), ciliary body (5–8%), and iris (3–5%), is the most common primary intraocular malignant tumor in adults [1]. Five-year overall survival (OS) across all stages is about 80% and has not changed in three decades [2,3]. Despite successful treatments of the primary tumor, up to 50% of patients with primary UM develop metastasis by hematogenous spread. The liver is the most predominant metastatic site of UM, with approximately 90% of patients metastasizing to the site [4,5,6]. Other common sites of metastasis include the lung at 29%, the bone at 17%, and skin/soft tissue at 12%. Once patients were diagnosed with metastatic uveal melanoma (MUM), 80% of them died within one year [4]. A clinical database in our institution demonstrates that the median survival of MUM patients improved from 5.3 months (1971–1993) to 17.8 months (2008–2017) with the shift of treatment modalities from DTIC-based chemotherapy to liver-directed therapies [7]. Unfortunately, MUM is almost always fatal.

Unlike cutaneous melanoma, which has driver mutations in v-raf murine sarcoma viral oncogene homolog B1 (*BRAF)*, neuroblastoma RAS viral oncogene homolog (*NRAS)*, and neurofibromatosis type 1 (*NF1)*, more than 90% of UM cases commonly harbor mutations in the alpha subunit of heterotrimeric G proteins, *GNAQ/GNA11*, that are involved in early UM development. These G proteins play an essential role in cellular signal transduction, and *GNAQ/GNA11* mutations drive carcinogenesis and cell proliferation [8,9,10]. The mutations of cysteinyl leukotriene receptor 2 (*CYSLTR2*) and phospholipase C beta 4 (*PLCB4*) were also found, and these lead to the activation of GNAQ/GNA11 signaling [11,12]. Secondary somatic alterations affect BRCA1 associated protein 1 (*BAP1*), splicing factor 3b subunit 1 (*SF3B1*), or eukaryotic translation initiator factor 1A X-linked (*EIF1AX*), and these mutations tend to correlate with the development of metastasis [10,13]. Inactivating somatic mutations were identified in *BAP1* on chromosome 3p21.1 in 26 of 31 (84%) metastasizing UM tumors and was correlated with the development of systemic metastasis and poor outcome [14]. Loss of 1 copy of chromosome 3 and *BAP1* aberrancy may result in a metastasis-prone DNA methylation state [15].

*SF3B1* mutations appear to be associated with the development of delayed metastasis, with a median duration of 8.2 years from diagnosis of primary UM to metastasis [16]. In contrast, *EIF1AX* mutations were rarely found in MUM [17]. In addition, serine- and arginine-rich splicing factor 2 (*SRSF2*) has also been found in 3 of 80 UM samples from The Cancer Genome Atlas (TCGA) and 2 of 42 UM tumors from the Rotterdam Ocular Melanoma Study group (ROMS) database [15,18]. Due to the small population of UM patients with *SRSF2* somatic mutation, the correlation of this mutation with the development of metastasis is unknown. Other research data indicated that 8q amplification in UM tumors was associated with the high risk of metastasis [13].

In non-malignant situations, hepatocyte growth factor (HGF)/mesenchymal-epithelial transition factor (MET) signaling is essential for embryogenesis, muscle development, nervous system formation, hematopoietic cell differentiation, and bone remodeling [19,20]. In malignant disease, the activation of the HGF/MET signal pathway results in cell proliferation, survival, inhibition of apoptosis, migration, invasion, and metastasis [21,22,23]. This is also true for UM. Previous publications indicated that HGF/MET signaling correlates with metastasis formation in UM. For example, Gardner et al. reported that MET expression was significantly higher in metastatic tissues than in primary tissues [24]. Barisione et al. reported that patients with metastatic disease had significantly higher serum levels of soluble c-Met [25]. Furthermore, Surriga et al. revealed that a MET inhibitor prevented liver metastasis in xenograft mouse models [26]. These data indicate that HGF/MET signaling may be involved in the formation of UM metastasis, and the inhibition of this signal pathway might be important in the treatment of MUM. In this review, we investigate how HGF/MET is involved with MUM and discuss the role of HGF/MET pathway inhibition as a treatment strategy for MUM.

## 2. Structure and Biological Function of Hepatocyte Growth Factor

HGF is a large multidomain protein that was discovered more than three decades ago. The gene-encoding *HGF* is located on chromosome 7q21.11 and consists of 18 exons and 17 introns. HGF was discovered as a mitogen of hepatocytes for liver regeneration. It was identified in the sera of 70%-hepatectomized rats as a mitogen of adult rat hepatocytes [19]. HGF is produced and secreted in the form of pro-HGF by hepatic stellate cells [27], fibroblast, and smooth muscle cells [28]. Pro-HGF is in its biologically inactive form. HGF is synthesized as a pro-HGF form of 728 amino acids comprising a single chain, including α- and β-subunits, and consisting of 6 domains: an N-terminal domain (N), four copies of the kringle domain (K1–K4), and a C-terminal domain (SPH) [29] (Figure 1a).

To activate pro-HGF, a single-chain HGF is cleaved between Arg494 and Val495 by three serine proteinases: the soluble HGF activator (HGFA), the type II transmembrane enzymes matriptase, and hepsin [20,30,31] (Figure 1a). HGFA is a soluble proteinase that is activated by thrombin, while matriptase and hepsin are expressed on the surface of MET-expressing target cells. Although α-subunit of both active-HGF and pro-HGF can bind to MET with high affinity, only active-HGF has signal transduction activity [29]. When cleavage occurs between Arg494 and Val495, α- and β-subunits are bound with a disulfide bond. It is speculated that the conformational change in N-terminal Val495 of SPH facilitates better functional binding to the MET molecule and activates MET signaling pathways [32]. Additionally, two fragment forms of HGF (NK1, NK2) occur naturally as splice variants. NK1 and NK2 consist of the N-domain and one (K1, NK1) or two (K1 and K2) kringle domains (NK2). Both NK1 and NK2 bind MET directly with high affinity. NK1 generally works as a MET agonist under the presence of heparin but could work as an antagonist without heparin. In contrast, NK2 has no agonistic activity but rather antagonizes HGF [33,34]. The mechanism of how HGF binding results in MET dimerization and signal transmission into cells remains unclear.

HGF is produced in various pathological events in the liver. For example, when 70% of the liver was removed, HGF mRNA levels markedly increased in intact distant organs, such as the lung, kidneys, and spleen. Pro-inflammatory cytokines such as interleukin (IL)-1, IL-6, interferon-gamma, and tumor necrosis factor-alpha (TNF-α) are involved in the upregulation of HGF gene expression in stroma cells. These pro-inflammatory cytokines trigger HGF production in intra-hepatic tissues via the paracrine loop and in extra-hepatic organs via the endocrine loop [19].

## 3. Structure and Biological Function of MET

Proto-oncogene *MET* codes a tyrosine kinase transmembrane receptor of HGF, composed of a 50-kDa α-subunit and 145-kDa β-subunit (Figure 1b). The proto-oncogene *MET* is located on chromosome 7q21-q31. MET has two domains, the extracellular and intracellular domains. The extracellular domain is composed of an α-subunit and a part of a β-subunit consisting of SEMA domain, plexin-semaphorin-integrin (PSI), and immunoglobulin-like fold-plexin-transcription factor (IPT) domain [35,36]. The α-subunit and SEMA domain of the β-subunit binds to HGF. The SEMA domain is necessary for receptor dimerization and activation [37]. The intracellular region of MET β-subunit comprises three segments: a juxtamembrane segment (JM), a tyrosine kinase (TK) domain, and a C-terminal docking site (Figure 1b). By binding HGF to MET, two subunits are dimerized, leading to the phosphorylation of Y1234 and Y1235 in the TK domain. Dimerization of MET is followed by phosphorylation of Y1349 and Y1356 in the C-terminal docking site, leading to the recruitment of adaptor proteins and growth factor receptor-bound protein 2 (GRB2). This then recruits multiple other proteins, including the docking protein Grb2-associated binder 1 (GAB1) and the CbI ubiquitin ligases, and signaling pathway-related molecules such as Son of Sevenless (SOS), Rous sarcoma oncogene cellular homolog (Src), Src homology 2 domain-containing (SHC), phosphatidylinositol 3 kinase (PI3K), signal transducer and activator of transcription 3 (STAT3), and others [35] (Figure 2). It is of note that the JM domain contains two protein phosphorylation sites, S985 and Y1003, and acts as a negative regulator. The Cbl ubiquitin ligase binds phosphorylated Y1003, and this Cbl binding results in MET ubiquitination, endocytosis, and subsequent degradation by the lysosome. S985 is phosphorylated by HGF-induced MET activation, and biological responses are suppressed [38]. Exon 14 of *MET* encodes the JM region, which contains key regulatory elements, including Y1003. *MET* dysregulation through splice-site alterations causes loss of transcription of exon 14, which leads to exon 14 skipping. MET activation with exon 14 skipping inhibits MET negative regulation, demonstrating uncoupled Cbl protein binding, decreased ubiquitination, and inefficient targeting for degradation of MET. Therefore, this exon 14 skipping enhances HGF-induced MET phosphorylation and prolongs MET activation. In this regard, cancer cells which have exon 14 skipping are sensitive to a MET inhibitor [39]. In addition to alterations, overexpression and gene amplification of *MET* is present in a variety of tumors and have been shown to correlate with poor prognosis [40,41]. HGF/MET signaling mediates the mitogen-activated protein kinase (MAPK)/extracellular signal-regulated kinase (ERK), PI3K/protein kinase B (AKT), focal adhesion kinase (FAK), and STAT3/5 signaling pathway (Figure 2). HGF and MET expression can be upregulated by basic fibroblast growth factor (b-FGF), TNF-α, IL-1, IL-6, and several other cytokines [28].

The complexity of MET signaling is a crosstalk with other receptors and membrane proteins. In complex multi-cellular organisms, the formation of a heterodimeric complex permits interaction and crosstalk between different receptors of the same subfamily. In some cases, HGF/MET signaling is mediated by other membrane proteins that respond to extracellular signals. A single (monomeric) MET interacts with these various cell-surface proteins. These membrane proteins include integrins, the proteoglycan CD44, G protein-coupled receptors (GPCRs), and other tyrosine kinase receptors such as insulin-like growth factor 1 receptor (IGF-1R), epidermal growth factor receptor (EGFR), and Erb-B2 receptor tyrosine kinase 3 (ERBB3) interact with a single (monomeric) MET and activate the downstream signal pathways [42]. Co-expression of MET and IGF-1R on primary UM tissue sections was reported as a prognostic value [43]. Ligand-independent activation of MET through IGF-1/IGF-1R signaling has also been reported [44].

## 4. The Role of HGF/MET Signaling in Metastatic Uveal Melanoma

HGF/MET signaling plays a key role in the development of metastasis in uveal melanoma (Figure 2 and Figure 3). Robertson et al. defined four subtypes from 80 UM TCGA samples and divided monosomy 3 (M3) and disomy 3 (D3) into two subgroups, according to clonal and subclonal somatic copy number alterations and whole-exome sequencing data. They indicated that M3-UM has a higher risk of metastasis and poorer prognosis than D3-UM [15]. In an analysis of 80 TCGA samples, MET RNA expression correlated to these 4 subtypes. MET RNA expression gradually increases to cluster 4, which is correlated to UM metastasis and poorer prognosis when compared with other clusters (Figure 4). This is also supported by our institutional data. In 28 metastatic uveal melanoma specimens, 24 (85.7%) showed positive MET protein expression in uveal melanoma cells (Table 1, Figure 5). Various mechanisms have been reported to support the importance of HGF/MET signaling in the progression of disease (Figure 3).

### 4.1. The Role of HGF/MET Signaling in Cell Migration and Invasion

Forming a metastatic lesion from a primary lesion involves multiple steps, including proliferation, invasion, migration, intravasation, dissemination, extravasation, and colonization in the distant organ. First, the cancer cells need to invade and migrate to adjacent tissues. Cell adhesion helps establish tight connections both between cells and cells and between cells and the matrix [45]. Therefore, loss of cell–cell and cell–matrix adhesion help tumor cell dissemination and metastasis. On the molecular level, cell–cell adhesion is mediated by two main groups of molecules: Ca-dependent family (cadherin, E-cadherin, N-cadherin, selectin, and integrin) and Ca-independent family (immunoglobulin and lymphocyte homing receptors) [46].

HGF/MET signaling controls cell migration and cell adhesion mainly through PI3K/AKT pathways in UM cell lines [47]. HGF also induces FAK activation in one of the downstream signaling pathways of MET. FAK plays an important role in integrin-initiated signaling pathways [48]. HGF and MET interaction leads to FAK activation and contributes to the regulation of cell–cell adhesion and cell–extracellular matrix (ECM) adhesion. FAK may be involved in HGF-induced cell motility, and that renders MET-expressed tumor cells susceptible to transformation by HGF stimulation to promote migration and invasion [49].

### 4.2. The Role of HGF/MET Signaling in Epithelial-Mesenchymal Transition

Epithelial-mesenchymal transition (EMT) is an essential step to migration, invasion, and extravasation from a primary lesion. The molecular hallmarks of EMT include E-cadherin down-regulation and up-regulation of mesenchymal-related proteins such as vimentin and N-cadherin to acquire mesenchymal-like characteristics allowing cells to move from the original site to a distant site [50]. Acquiring EMT characteristics, several million cells per gram of tumor can be shed daily into the lymphatic system or into the bloodstream while circulating cancer cells (CTC) have to overcome anoikis to achieve metastasis [46].

Several studies indicated that the HGF/MET signaling pathway promotes EMT through up-regulation of the E-cadherin repressor and mediates the switch from E-cadherin to N-cadherin in cutaneous melanoma cell lines [51,52]. Li et al. reported that HGF induced the downregulation of E-cadherin and Desmoglein 1, which are required to induce cell scattering. HGF-induced E-cadherin and Desmoglein 1 downregulation depend on the MAPK and PI3K pathway [52]. In UM, reduced E-cadherin expression in primary tumors was reported to be correlated to shorter survival. The expression level of E-cadherin mRNA was lower in metastatic tumors than in primary tumors [53].

It is of note that the E-cadherin expression and epithelial-mesenchymal transition in UM are controversial. Harbour et al. indicated that the up-regulation of CDH1 and membranous E-cadherin expression in primary UMs are associated with class 2 characteristics, having a high metastatic risk [54,55]. In fact, UM with spindle cells showed a favorable prognosis, whereas poor prognosis was seen in patients with an increasing number of epithelioid cells [56]. This discrepancy may be explained by two reasons. One is UM does not arise from an epithelium, the other is E-cadherin has different roles between early-stage and late-stage in tumor progression [57]. Loss of E-cadherin induces metastasis, but E-cadherin is often re-expressed in metastatic lesions [58]. Further studies are needed to investigate the association between E-cadherin and metastasis in UM.

### 4.3. The Role of HGF/MET Signaling in Survival in the Bloodstream

In general, cells stay close to the tissue to which they belong since the communication between neighboring cells as well as between cells and ECM provides essential signals for growth or survival. When cells are detached from the ECM, there is a loss of normal cell–matrix interactions, and they may undergo anoikis. Therefore, the escape mechanism from anoikis is important for tumor cell survival in the bloodstream and different organs. It has been reported that anoikis resistance can be induced through HGF activating both extracellular signaling-receptor kinase (ERK) and PI3K [59].

The PI3K/AKT/mTOR (mechanistic targeting of rapamycin) pathway that is activated by HGF/MET signaling is one of the most important pathways involved in pro-survival features, as it integrates most of the signals derived from integrins and growth factor receptors. This pathway is essential to regulate several cellular functions, such as cell survival and cell growth. AKT activation can modulate the activity of transcription factors that stimulate anti-apoptotic genes or directly phosphorylate pro-apoptotic proteins, such as Beclin (Bcl)-2 antagonist of cell death (Bad) and procaspase-9, inhibiting their function [60]. The activation of AKT subsequently inhibits apoptosis by activating the E3 ubiquitin-protein ligase MDM2, an inhibitor of p53. In addition, Glycogen synthase kinase 3 beta (GSK3β), downstream of AKT, is phosphorylated, which results in its inhibition, leading to p53 inhibition and subsequently protects apoptosis.

### 4.4. The Role of HGF/MET Signaling in Angiogenesis

HGF/MET signaling is also a potent inducer of endothelial cell growth and promotes angiogenesis and lymphangiogenesis, in addition to vascular endothelium growth factor (VEGF) and b-FGF signaling [61]. Furthermore, HGF/MET signaling can induce vascular endothelium growth factor A (VEGFA) expression and angiogenesis through common signaling pathways, such as SHC. Thrombospondin 1 (TSP1; also known as THBS1) is a negative regulator of angiogenesis that is suppressed by HGF/MET signaling. By regulating VEGFA and TSP1 expression, HGF/MET signaling acts as a potent regulator of angiogenesis [62]. Induction of angiogenesis by HGF supplementation resulted in improved local hypoxia.

### 4.5. The Role of HGF/MET Signaling in Production of Metalloproteinase

Koh and Lee showed that HGF upregulates matrix metalloproteinase-9 (MMP-9) in 2 metastatic gastric carcinoma cell lines [63]. MMP-9 is a member of MMPs that break down the basement membranes through the degradation of type IV collagen, exposing cryptic sites within the matrix and allowing cancer cell invasion. Degradation of ECM in the tissue of the tumor is a principal process of cancer invasion and metastasis [63]. MMP-9 is particularly correlated with pro-oncogenic events such as neo-angiogenesis, tumor cell proliferation, and metastasis [46]. MMP-9 is shown to play an important role in tumor dissemination. The value of MMP-9 is evaluated as a biomarker for various specific cancers. In UM, the expression of MMP-2 and MMP-9 has been associated with a higher incidence of metastasis. MMP-9 was predominantly present in epithelioid UM and the epithelioid portion of mixed-cell UM [64].

## 5. Targeting HGF/MET Signaling in Metastatic Uveal Melanoma

To understand the structure–function relationship of the ligand, cancer therapy targeting HGF/MET signaling has been developed [65]. There are many types of drugs that inhibit HGF/MET signaling. There are three types of inhibitors, including hepatocyte growth factor activator (HGFA) inhibitor, MET antagonist, and MET signal inhibitor for inhibition of MET signaling activity. HGFA inhibitor (HAI)-1 inhibits serine proteases including hepsin, matriptase, HGFA, and proteasin, blocking the activation of pro-HGF [66]. The MET antagonist binds to the SEMA domain of MET, acting as a potent agonist [67]. NK4, as an HGF antagonist, can inhibit tumor invasion, growth, angiogenesis, and metastasis of tumors in vivo [68].

There are many different types of MET signal inhibitors developed over the past decade and have been investigated in ongoing clinical studies. Type I MET inhibitors preferentially bind to the active confirmation of MET kinase, while type II MET inhibitors bind to the inactive confirmation [69]. Crizotinib (Xalkori), a type I MET inhibitor, is a small-molecule multi-kinase inhibitor. Crizotinib was approved for metastatic anaplastic lymphoma kinase (ALK)-positive non-small cell lung cancer (NSCLC) in 2011 and metastatic ROS proto-oncogene 1 receptor tyrosine kinase (ROS1)-rearranged NSCLC in 2016 [70]. MET kinase domain mutations have emerged as mechanisms of resistance to crizotinib in patients with *MET*-amplified and *MET* exon-14-altered cancers. Crizotinib was not effective as adjuvant therapy for patients with high-risk UM (NCT02223819). Currently, a phase 2 study of crizotinib or binimetinib with protein kinase C (PKC) inhibitor in patients with solid tumors harboring *GNAQ/GNA11* mutations or protein kinase C (PRKC) fusion is ongoing (NCT03947385).

Merestinib and cabozantinib, type II MET inhibitors, have preclinical activity against several kinase domain mutations (such as D1228N, M1250T, and H1094Y/L145). Merestinib (LY2801653) is an oral kinase inhibitor with anti-proliferative and anti-angiogenic activity in *MET*-amplified and MET autocrine xenograft tumor models [71,72]. Merestinib is also active against other receptor tyrosine kinase and serine/threonine kinases (MKNK1 and MKNK2) [70]. Ohara et al. demonstrated that merestinib enhances the effect of cyclin-dependent kinase (CDK)4/6 inhibitor for MUM [73]. Currently, a phase 2 study of merestinib or ramucirumab with cisplatin and gemcitabine combination in patients with advanced or metastatic cancer is ongoing (NCT 0271155) [74].

Cabozantinib is a multi-kinase inhibitor that inhibits MET and vascular endothelium growth factor receptor 2 (VEGFR2) phosphorylation. Cabozantinib, therefore, inhibits the migration, invasion, and proliferation in a dose-dependent manner [70]. Cabometyx (cabozantinib) is approved for metastatic medullary thyroid cancer, hepatocellular carcinoma, and renal cell carcinoma (RCC). A phase 2 study comparing cabozantinib with temozolomide or dacarbazine for MUM patients has been conducted (NCT01835145) [75].

Additionally, Tabrecta (capmatinib) is approved for patients with metastatic NSCLC with *MET* exon 14 skipping (METex14).

## 6. The Role of HGF/MET Signaling in Therapeutic Resistance

Various treatment approaches have been tested for MUM using targeted therapy and immune checkpoint inhibitors. However, the efficacy of these therapies is limited due to pre-existing and/or acquired therapeutic resistance mechanisms. Especially, the tumor microenvironment plays an important role in resistance to cancer therapeutics, including chemotherapy, radiation therapy, targeted therapy, and immunotherapy. In particular, HGF produced by hepatic stellate cells and stromal cells is known to induce resistance to cancer therapy. In this regard, the majority of currently available mouse systems might not be suitable to investigate the role of HGF/MET signaling in cancer treatment resistance mechanisms since mouse HGF has limited binding affinity to human MET [76]. Potentially efficacious treatment approaches developed in traditional mouse models might not be applicable to patients since the influence of human HGF in the MUM microenvironment is not included in these mouse models.

### 6.1. Resistance to Chemotherapy

HGF/MET signaling induces chemotherapy resistance [77]. Xu et al. revealed that HGF/MET signaling enhances gemcitabine chemoresistance in pancreatic cancer cell lines [78]. Chen et al. reported that HGF suppresses N-methyl-d-aspartate-induced apoptosis-inducing factor (AIF) through activation of FAK and induces cisplatin resistance in lung cancer cells [79]. HGF/MET signaling inhibits AIF and protects the apoptotic cell death followed by DNA damage [80]. Canadas et al. revealed that MET inhibitor reverses mesenchymal transition and increases chemosensitivity in small cell lung cancer [81]. Although chemotherapies have not recently been used for MUM, HGF/MET could be targeted to overcome chemoresistance and improve overall survival.

### 6.2. Resistance to Targeted Therapy

More than 90% of UM cells harbor *GNAQ/GNA11* mutations; therefore, targeted therapy against this pathway might be effective in UM. Since *GNAQ/GNA11* mutations are not easily targetable, research has focused on downstream pathways, such as MAPK, PKC, PI3K, and AKT signaling. Despite impressive growth suppression in in vitro and preclinical mouse models, these signal inhibitors only showed marginal effects on patients with MUM. For example, selumetinib, a mitogen-activated protein kinase kinase (MEK)1/2 inhibitor, did not improve overall survival in MUM patients [82]. Clinical trials using selumetinib in combination with dacarbazine did not show a survival benefit compared to chemotherapy alone [83]. Furthermore, a different MEK inhibitor, trametinib, with or without GSK795 of AKT did not yield clinical benefit [84]. These clinical trials indicated that one of the reasons for failure is the existing resistance mechanisms to targeted therapies. In this regard, it is highly likely that MUM has a primary resistance mechanism to MEK inhibitors that HGF provides. Cheng et al. revealed that paracrine effects of HGF from fibroblasts protect UM cells from MEK inhibition. Targeting HGF/MET signaling can overcome the resistance elicited by HGF [85,86]. We previously reported that HGF mediates resistance to CDK4/6 inhibitor in MUM through activation of the HGF/MET signaling pathway. Dual targeting CDK4/6 and MET overcomes the resistance and might be more effective than CDK4/6 monotherapy [73]. It is also known that HGF induces resistance to EGFR-targeted therapy due to crosstalk between MET and EGFR, regardless of wild-type or mutant EGFR [80]. Furthermore, HGF/MET promotes angiogenesis via upregulation of VEGFA and suppression of TSP-1. The combination of MET and VEGFR inhibitors demonstrated strong inhibition of tumor growth and tumor angiogenesis in xenograft models [87].

It is also true that treatments with single-agent HGF/MET signaling inhibitors failed to show dramatic improvement in the treatment of cancer patients. It has been reported that autophagy, one of several cellular adaptive responses to therapeutic stresses caused by anti-cancer agents, is critical for the resistance to HGF/MET-targeted therapy [88]. The mechanism that regulates HGF/MET-targeted drug resistance is quite complicated, and further investigation is required.

### 6.3. Resistance to Radiation Therapy

Radiation therapy causes DNA damage and induces apoptotic cell death [89]. Therefore, activation of anti-apoptotic signals such as the PI3K/AKT pathway induces resistance to radiation therapy. Several studies have shown the stimulation of HGF/MET signaling by radiation therapy. In neuroblastoma, radiation enhances HGF mRNA expression and MET amplification [90]. Raghav et al. indicated radiation therapy-induced MET overexpression, which is a key role in the development of resistance to radiation therapy [80]. In addition, hypoxia-induced radiation therapy can cause transcriptional activation of *MET* proto-oncogene [91].

### 6.4. Resistance to Immunotherapy

HGF/MET signaling is also involved in immune responses in the tumor microenvironment, potentially having a protumor effect. HGF/MET signaling can recruit neutrophils to the tumor microenvironment from the bone marrow [35]. MET-expressing neutrophils reduce T cell proliferation in the tumor microenvironment. Furthermore, in lung cancer, MET expression and gene amplification are correlated with programmed cell death ligand 1 (PD-L1) expression in cancer cells [92,93]. It has been reported that HGF/MET signaling directly induces PD-L1 expression [92]. In fact, cabozantinib combination with an immune checkpoint drug has significant benefits and is approved for advanced renal cell carcinoma. These results suggest that MET inhibitor co-treatment may improve responses to cancer immunotherapy by activating T cell-mediated anti-cancer immunity.

On the other hand, there are controversial data suggesting that MET/HGF signaling is also considered to induce an antitumor effect. MET is required for chemoattraction and cytotoxicity of neutrophils in response to its ligand, HGF. HGF/MET-induced neutrophils activation produces nitric oxide that kills tumor cells directly. MET deletion in neutrophils leads to the enhancement of tumor growth and metastasis [94]. There are multiple hypotheses around the lack of efficacy with checkpoint inhibitor therapies in MUM [95], and additional studies are needed to investigate the role of HGF/MET signaling in the immune response against MUM.

## 7. Conclusions

HGF/MET signaling plays an important role in MUM through various mechanisms. It is also involved in not only metastatic formation but also in resistance to other therapies. Clinical trials targeting HGF/MET signaling are ongoing in patients with MUM. These trials may lead to improvement of overall survival in MUM through an impactful synergy effect with another therapy and by overcoming therapeutic resistance.

## Figures and Tables

**Figure 1 cancers-13-05457-f001:**
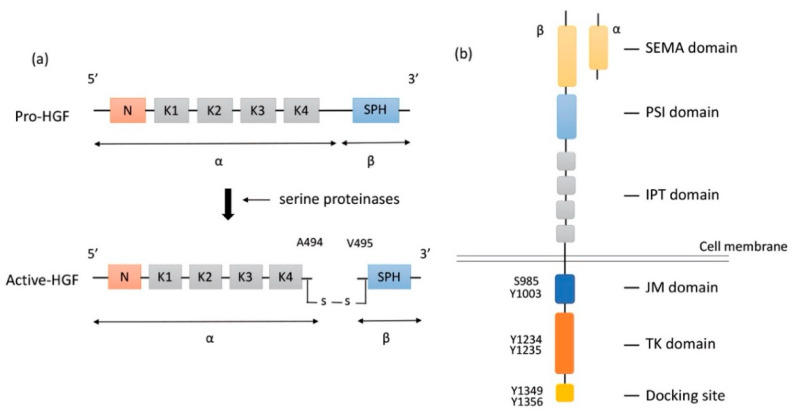
Structure of HGF and MET. (**a**) Structure of HGF; Secreted pro-HGF form a single chain, consisting of six domains: an N-terminal domain, four copies of the kringle domain (K1–K4), and a C-terminal domain (SPH). Pro-HGF is cleaved between Arg494 and Val495 by serine proteinases. In active HGF, α- and β-subunits are bound with a disulfide bond. (**b**) Structure of MET; HGF binds to SEMA domain of MET, followed by dimerization and activation. Y1234 and Y1235 in the TK domain are phosphorylated, and then phosphorylation of Y1349 and Y1356 in the C-terminal docking site leads to the recruitment of adaptor proteins and signaling molecules. HGF: hepatocyte growth factor; MET: mesenchymal epithelial transition; PSI: plexin-semaphorin-integrin; IPT: immunoglobulin-like fold-plexin-transcription factor; JM: juxtamembrane; TK: tyrosine kinase.

**Figure 2 cancers-13-05457-f002:**
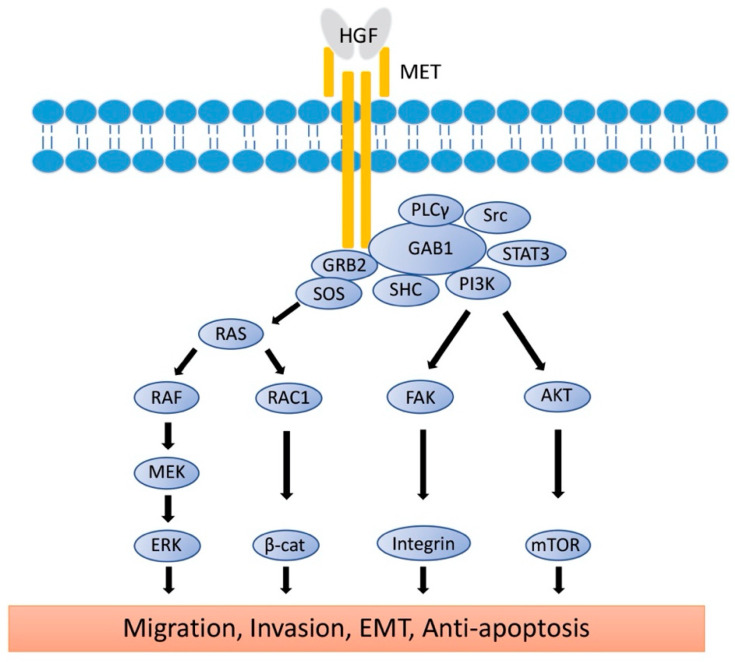
Signaling Pathways of HGF/MET. The active form of HGF induces the dimerization of MET and activates the signaling pathway. Phosphorylation of MET leads to the recruitment of GRB2 to the C-terminal docking site, followed by binding of GAB1, SOS, Src, SHC, PI3K, and STAT3. RAS/MAPK and PI3K/AKT/mTOR signaling molecules reach the nucleus to induce cell proliferation, EMT, and anti-apoptosis. Rac1/β-catenine and FAK/Integrin signaling pathways reach the cell membrane to control E-cadherin and integrin expression for migration, invasion, and EMT. HGF: hepatocyte growth factor; MET: mesenchymal epithelial transition; GRB2: Growth factor receptor-bound protein 2; GAB1: Grb2-associated binder 1; SOS: son of sevenless; Src: rous sarcoma oncogene cellular homolog; PLCγ: phosphoinositide phospholipase C γ; SHC: Src homology 2 domain-containing; PI3K: phosphatidylinositol 3 kinase; mTOR: mechanistic target of rapamycin; STAT3: signal transducer and activator of transcription 3; AKT: protein kinase B; RAS: rat sarcoma virus; RAF: rapidly accelerated fibrosarcome; RAC1: ras-reated C3 botulinum toxin substrate 1; β-cat: β-catenin; FAK: focal adhesion kinase; MEK: mitogen-activated protein kinase kinase; ERK: extracellular signal-regulated kinases; MAPK: mitogen-activated protein kinases; EMT: epithelial-to-mesenchymal transition.

**Figure 3 cancers-13-05457-f003:**
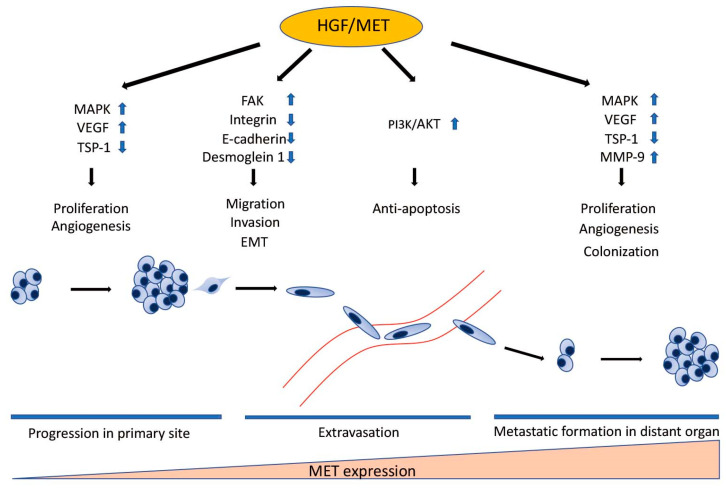
The Role of HGF/MET in Individual Steps of Metastasis. HGF/MET signaling regulates cell proliferation, migration, invasion, EMT, survival in the bloodstream, and colonization in distant organs through many signaling pathways. MET expression correlates to the development of metastasis. MET RNA expression gradually increases in proportion to the step of the metastatic process. HGF: hepatocyte growth factor; MET: mesenchymal epithelial transition; MAPK: mitogen-activated protein kinases; VEGF: vascular endothelium growth factor; TSP-1: thrombospondin 1; FAK: focal adhesion kinase; EMT: epithelial-to-mesenchymal transition; PI3K: phosphatidylinositol 3 kinase; AKT: protein kinase B; MMP-9: matrix metalloproteinase-9.

**Figure 4 cancers-13-05457-f004:**
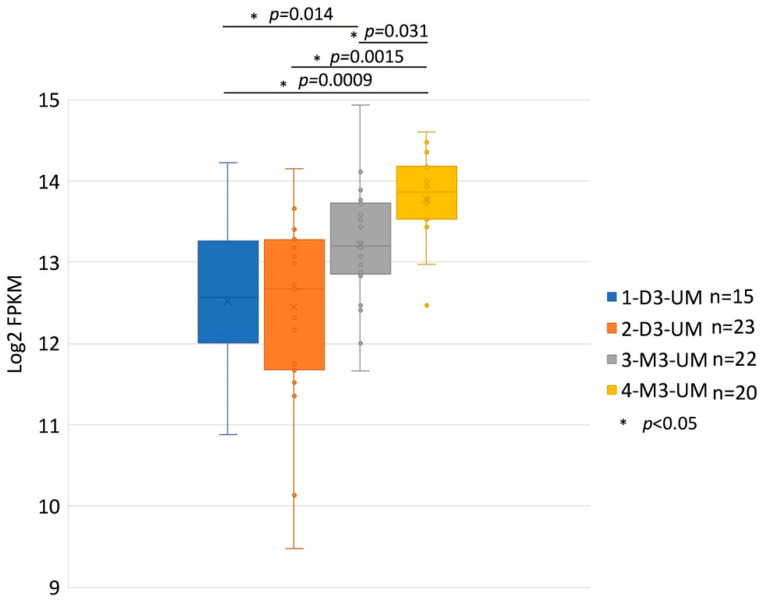
MET RNA Expression within the Four Somatic Copy Number Alterations (SCNA) Cluster Groups. These data are derived from 80 primary UM samples in the TCGA database and were stratified SCNA clustering defined 4 subtypes. The y-axis is a log2 with FPKM values of MET RNA. Box plots show median values and the 25th to 75th percentile range in the data. 1-D3-UM vs. 2-D3-UM *p* = 0.831; 1-D3-UM vs. 3-M3-UM *p* = 0.014; 1-D3-UM vs. 4-M3-UM *p* = 0.0009; 2-D3-UM vs. 4-M3-UM *p* = 0.0015; and 3-M3-UM vs. 4-M3-UM *p* = 0.031. MET: mesenchymal epithelial transition; UM: uveal melanoma; TCGA: The Cancer Genome At-las; D3: disomy 3; M3: monosomy 3.

**Figure 5 cancers-13-05457-f005:**
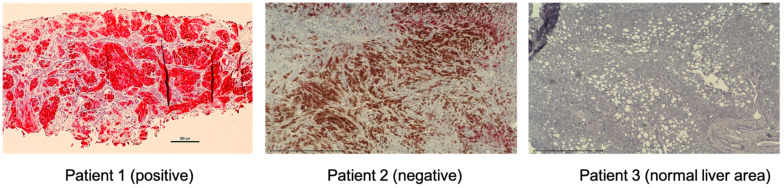
MET expression of UM metastatic tumor samples by immunohistochemistry. Representative images of positive and negative MET expression and normal liver area are shown (magnification, ×200). Tumor biopsy samples from UM patients were stained with CONFIRM anti-Total c-MET (SP44) rabbit monoclonal primary antibody (Ventana, Medical Systems). These specimens from metastatic uveal melanoma patients were re-trieved in paraffin-embedded archival core biopsy for staining. The fast red chromogen (Ventana, Medical Systems) was applied for color development. The staining intensity was assigned a score of 0–3 (0 = no staining, 1 = weak, 2 = moderate, 3 = strong). The criteria for positive results; Intensity score ≥2 and ≥50% of cells stained. MET-expressing cells were stained red. The brown color on patient 2 is from pigmented melanoma cells. MET: mes-enchymal epithelial transition; UM: uveal melanoma.

**Table 1 cancers-13-05457-t001:** Expression of MET in Metastatic Uveal Melanoma Tissues of 28 Patients.

MET Positive		MET Negative	
Location	Number	Location	Number
Liver	21	Liver	2
Lung	1	Omentum	1
Abdomen	1	Mediastium	1
Cervical Lymph node	1		

## Data Availability

Publicly available dataset was analyzed in this study. This data can be found here: [https://portal.gdc.cancer.gov/projects/TCGA-UVM].

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
