# Peer review of "The Role of HGF/MET Signaling in Metastatic Uveal Melanoma"

_cancers, 2021, doi:10.3390/cancers13215457_

Round 1

Reviewer 1 Report

The role of HGF/MET in uveal melanoma has been somewhat understudied and thus this review attempts to bring together the key data for this signalling pathway in the field of uveal melanoma as well as an overview of its function of in cancer.  

Because of the lack of research in this area in uveal melanoma, the points being made about HGF/MET in this disease seem a little lost in each of the paragraphs i.e. there is a lot of information about HGF/MET with little reference back to how this could impact uveal melanoma or may be used to advance the field. The authors should try to highlight this and bring the HGF/MET work together as a whole in MUM.

The authors discuss MET expression and show data in Table 1. Is this protein expression by immunohistochemistry? If so, images of the staining and how the slides were assessed would be useful.

Reviewer 2 Report

This manuscript reviews HGF/MET signaling in metastatic uveal melanoma and is partially based on the author’s previous experiments with inhibition of this pathway in vitro in metastatic uveal melanoma cell lines, as well as in mice models with human HGF expression and in patients ’s derived tumor explants.

This review is interesting and well-constructed.

It would be valuable to illustrate in metastatic uveal melanoma as well as in the local microenvironment the expression of cMET (and pcMet if possible) as well as HGF. The authors previously demonstrated that media derived from early passage hepatic stellate cells induced migration and invasion of metastatic uveal melanoma cell lines and contained HGF. In that sense, the local tissue distribution of HGF and MET could be interesting and might nicely complement figure 3 and table 1.

4.1 The role of HGF/ MET signaling in cell migration: In my opinion, the authors should mention here their previous work with migration of UM metastatic cell lines. They could also mention the following work: Investigative Ophthalmology & Visual Science February 2008, Vol.49, 497-504.

4.2 in references 50, the authors did not use melanoma cells lines, but HepG2 and MDCK. For reference 51, cutaneous should be added to melanoma.

4.5. Page 9, line 280, I would add after MMP-9 “in 2 metastatic gastric carcinoma cell lines”.
